# Molecular Mechanisms of Cellular Injury and Role of Toxic Heavy Metals in Chronic Kidney Disease

**DOI:** 10.3390/ijms231911105

**Published:** 2022-09-21

**Authors:** Manish Mishra, Larry Nichols, Aditi A. Dave, Elizabeth H Pittman, John P. Cheek, Anasalea J. V. Caroland, Purva Lotwala, James Drummond, Christy C. Bridges

**Affiliations:** 1Department of Biomedical Sciences, Mercer University School of Medicine, Macon, GA 31207, USA; 2Department of Pathology and Clinical Sciences Education, Mercer University School of Medicine, Macon, GA 31207, USA

**Keywords:** chronic kidney disease, heavy metals, cadmium, mercury, arsenic, cellular injury

## Abstract

Chronic kidney disease (CKD) is a progressive disease that affects millions of adults every year. Major risk factors include diabetes, hypertension, and obesity, which affect millions of adults worldwide. CKD is characterized by cellular injury followed by permanent loss of functional nephrons. As injured cells die and nephrons become sclerotic, remaining healthy nephrons attempt to compensate by undergoing various structural, molecular, and functional changes. While these changes are designed to maintain appropriate renal function, they may lead to additional cellular injury and progression of disease. As CKD progresses and filtration decreases, the ability to eliminate metabolic wastes and environmental toxicants declines. The inability to eliminate environmental toxicants such as arsenic, cadmium, and mercury may contribute to cellular injury and enhance the progression of CKD. The present review describes major molecular alterations that contribute to the pathogenesis of CKD and the effects of arsenic, cadmium, and mercury on the progression of CKD.

## 1. Introduction

Chronic kidney disease (CKD) is a major public health concern. According to the Centers for Disease Control and Prevention (CDC), about 15% of adults in the United States, or 37 million, are estimated to have some degree of CKD [1,2]. The Global Burden of Disease study from 2017 reported 697.5 million cases worldwide. It is clear that CKD is an important contributor to morbidity and mortality. More than 2.5 million patients are receiving renal replacement therapy and that number is expected to double to 5.4 million by 2030 [3,4].

CKD is a deteriorating, progressive, and irreversible loss of renal function. It is characterized by the presence of structural and/or functional abnormalities of the kidney with associated health implications that last more than three months. Individuals with CKD may have albuminuria, urine sediment abnormalities, abnormal renal imaging findings, serum electrolyte or acid-base derangements, and an estimated glomerular filtration rate (eGFR) of <60 mL/minute/1.73 m^2^ [2,5]. CKD can progress silently to the advanced stages before the patient is aware of the disease; therefore, early detection and diagnosis are critical to slow or prevent progression [2,3]. There are different patterns of renal decline in patients with CKD. These patterns can be classified into very fast, fast, moderate, or slow decline, depending on the rate at which renal function declines. The differences in these patterns of decline reflect the heterogeneity of CKD origins and the related pathologies, adjunct comorbidities, and other harsh environmental exposures [6,7]. Diabetes mellitus, hypertension, obesity, and older age are the primary risk factors, while other risk factors include cardiovascular disease and a family history of CKD [2,3]. In addition, other factors such as human immunodeficiency virus (HIV) and exposure to toxicants or heavy metals have been shown to contribute to the development of CKD [8,9].

## 2. Pathophysiology of Chronic Kidney Disease

CKD can be due to various pathologic mechanisms, which can injure various components of the kidney, e.g., vasculature, tubular epithelial cells, interstitium, or the glomerulus. CKD is a consequence of two interrelated issues: an initial catalyst and a prolonging mechanism. The initial catalyst can be an intrinsic kidney defect, an inflammatory or immune-mediated cause, or exposure to nephrotoxicants [10]. The initial problem causes injury of affected glomeruli and leads to compensatory mechanisms in healthy nephrons, including glomerular hypertrophy and hyperfiltration. A continued cycle of hyperfiltration and consequent injury leads to sclerosis and nephron loss [11]. Although hyperfiltration initially compensates (partially) for the loss of functioning nephrons, the vicious cycle of nephron hypertrophy and sclerosis is a continuous process that results in CKD [12,13]. Atrophy and sclerosis of nephrons lead to further renal decline and alterations in normal fluid and solute homeostasis [10,11,12].

The purpose of this review is to describe some of the structural and molecular mechanisms that are involved in the pathogenesis of CKD (Figure 1). The pathogenic mechanisms may occur simultaneously, may be consequences of initial injury, or may be due to compensatory processes following injury. While this review covers a number of major cellular and molecular mechanisms that are involved in the pathogenesis of CKD, it should be recognized that there are numerous other factors that play a role in the pathogenesis and progression of CKD. Due to the complexity of the disease, is not possible to include all of the factors that contribute to its pathogenesis in one review.

### 2.1. Structural Alterations within the Kidney

#### 2.1.1. Podocytes

Podocytes are terminally differentiated cells that have three distinct compartments: the cell body, primary processes, and foot processes. Foot processes from adjacent podocytes are connected by a slit diaphragm, which forms the most selective component of the glomerular filtration barrier. Hyperfiltration causes foot processes and the slit diagram to be exposed to high tensile and fluid flow shear stress, which can cause stretching, injury, and loss of barrier function [14,15]. The changes in hemodynamic forces within the glomerulus modify the actin cytoskeleton of podocyte foot process, which can lead to effacement and disruption of the filtration barrier [16,17,18,19,20]. If hyperfiltration is not corrected, podocytes may detach, which leads to podocyturia, proteinuria, and decreases in GFR. Detachment is irreversible and often leads to glomerulosclerosis and CKD [21]. Studies in mice have found that vascular endothelial growth factor (VEGF)-A is required in developing podocytes to establish and maintain the glomerular filtration barrier [22]. In addition, the soluble VEGF receptor 1 (sFlt-1) appears to play a role in endothelial dysfunction observed in CKD [23]. Loss of VEGF-A can lead to dysregulation of podocytes, loss of endothelial cells, and collapse of glomerular capillaries [22].

#### 2.1.2. Capillary Network

Peritubular capillary rarefaction (loss of capillary density) is a common feature of CKD. Interestingly, capillary rarefaction correlates strongly with CKD and has been used as a predictor of progression. Loss of peritubular capillaries is strongly associated with hypoxia and interstitial fibrosis, which lead to renal functional decline [24,25,26]. Alterations in the expression of endothelial cell-derived factors such as VEGF-A, angiopoietin, and thrombospondin-1 lead to an imbalance of proangiogenic and antiangiogenic factors. Thrombospondin-1 is reported to inhibit renal tubular epithelial cell proliferation, due to reperfusion injury, via the CD47 receptor [27]. Furthermore, inflammatory cytokines such as interleukin (IL)-1α and tumor necrosis factor-α (TNF-α) are secreted and block VEGF-A expression, a major proangiogenic factor [28]. It has been reported that small arterial changes might be crucial primary contributors to the development of glomerulosclerosis due to the decreased number of pericapillary pericytes, as pericytes are crucial for peritubular vessel function and capillary survival [29,30]. Thus, the alteration or disequilibrium of nitric oxide, endothelin-1, endostatin, throbospondin-1, and VEGF might be causative for a functional disruption in capillaries, leading to chronic hypoperfusion, ischemia, and nephron loss.

#### 2.1.3. Tubular Epithelial Cells

The renal proximal tubule absorbs the majority of filtered solutes in an energy-consuming process. The reported contribution of acute kidney injury (AKI) to CKD has focused attention on the proximal tubule as the main target of injury in the progression of CKD. Cells with high energy demand and slow proliferation, such as proximal renal tubules, are predisposed to oxidative damage and consequent injury [31]. Juvenile mice with a targeted deletion of endothelial nitric oxide synthase (eNOS) display renal cell death and renal cortical scars [32]. It is reported that patients with eNOS polymorphisms have increased susceptibility to the progression of CKD and thus, eNOS has been recognized as an important survival factor [33]. Tubular cells, which have a long life and high metabolic activity, rely heavily on proper mitochondrial function [34]. Mitochondrial density remains constant in hypertrophied renal tubular cells, but mitochondrial volume increases by over 50% [35]. Increased functionality may lead to additional mitochondrial oxidative stress, resulting in mitochondrial dysfunction. This dysfunction may lead to inflammation, additional alterations in intracellular homeostasis, and additional cellular injury [31,36,37].

### 2.2. Intracellular Alterations

#### 2.2.1. Mitochondrial Function

Human kidneys comprise only 1% of body weight, but they utilize approximately 10% of total body oxygen. After loss of functional renal mass, nephrons become hypertrophic, which increases the oxygen consumption by up to 50% [38]. This level of energy expenditure cannot be maintained indefinitely and is limited by the capacity of mitochondria to match increased demands [34]. The means by which renal cells produce ATP varies among cell type. Proximal tubular cells produce ATP via oxidative phosphorylation, while podocytes, endothelial, and mesangial cells utilize glycolysis. These differences may determine the impact of mitochondrial dysfunction in renal cells and affect the progression of renal diseases [36]. It has been demonstrated that acute or chronic insults cause mitochondrial structural alterations, including mitochondrial DNA (mtDNA) damage and reduced matrix density [37]. The main mitochondrial dysfunction includes altered mitochondrial biogenesis, fusion/fission, mitophagy, and impaired homeostasis, which lead to a decrease in ATP production, alterations in calcium signaling, enhanced oxidative stress, and apoptosis [36,37].

Mitochondrial biogenesis is regulated primarily by peroxisome proliferator-activated receptor γ coactivator-1 (PGC-1α) [37]. PGC-1α activates many transcription factors such as nuclear respiratory factor-1 (NRF-1), NRF-2, and the estrogen-related receptors (ERR) [39], and together, these transcription factors regulate genes involved in mitochondrial biogenesis, lipid oxidation, glycolysis, and ATP biosynthesis. Reduced expression of PGC-1α along with decreased efficiency of mitochondrial biogenesis has been observed in CKD [36,40,41]. It is not clear if this reduced expression is a cause or an effect of CKD.

Alterations in mitophagy have been implicated in several kidney diseases, including CKD [42,43]. The expression of BNIP3 (Bcl2-interacting protein 3), a member of the Bcl2 (B cell lymphoma 2) family involved in mitophagy, is strongly reduced in renal tubular cells from diseased kidneys. This suggests that a disruption of mitochondrial quality control contributes to the pathogenesis of CKD [44]. Increased mitochondrial reactive oxygen species (ROS) causes inflammation and mitochondrial genome mutations, leading to mitochondrial dysfunction, which further increases ROS production and contributes to more mtDNA damage. This progressive damage in the mitochondrial genome has been implicated as a factor in the pathogenesis and acceleration of CKD [37,45].

#### 2.2.2. Oxidative Stress

Oxidative stress occurs when there is a disruption in the balance of free radicals and antioxidants that degrade those free radicals [46]. ROS and reactive nitrogen species (RNS) are generated during oxidative phosphorylation [47]. Under normal conditions, moderate concentrations of ROS/RNS act as second messengers that regulate signal transduction pathways [48]. Mitochondrial dysregulation increases the production of ROS beyond the normal levels, which depletes antioxidants and leads to oxidative stress [46]. Increased ROS leads to lipid, DNA, and protein oxidation, which cause the formation of complex radical intermediates [46]. These highly reactive intermediates trigger the secretion of pro-inflammatory mediators during active inflammation [49]. Inflammatory cytokines, such as IL-6 and TNF-α, promote renal injury by perpetuating dysregulation in cellular processes [47]. Therefore, oxidative stress exacerbates cellular damage and enhances the progression of CKD [46].

#### 2.2.3. Autophagy

Autophagy is a catabolic process that helps cells remove endogenous waste material and recycle cellular components [50]. Thus, autophagy is an essential pro-survival mechanism during cellular stress. Autophagy is initiated due to activation of major nutrient-sensing pathways, such as mammalian target of rapamycin complex-1 (mTOR1), adenosine monophosphate-activated protein kinase (AMPK), and sirtuin 1 [50]. Ischemic, toxic, immunological, and oxidative injury can enhance autophagy in proximal tubular cells and podocytes, modifying the course of renal diseases [50,51]. Dysregulation of autophagy can lead to progressive deterioration of renal function due to the accumulation of intracellular damaged proteins and enhanced oxidative stress [51]. Thus, autophagy is important in podocytes and proximal tubular cells to maintain proper homeostasis and prevent cellular injury [51,52]. Indeed, studies in mice and rats reported that a reduction in autophagy was associated with a buildup of dysfunctional mitochondria [53] and apoptosis [52,54].

#### 2.2.4. Endoplasmic Reticulum Stress

There is experimental evidence that the accumulation of unfolded proteins in the endoplasmic reticulum of renal tubular epithelial cells may play a role in their death and associated interstitial fibrosis [55]. This accumulation causes endoplasmic reticulum stress, thereby activating the unfolded protein response, which is an evolutionarily conserved cellular response primarily regulated by the endoplasmic reticulum-resident chaperone glucose-regulated protein 78 (GRP78). Misfolded proteins cause GRP78 to dissociate from endoplasmic reticulum transmembrane proteins (PKR-like endoplasmic reticulum kinase, inositol-required enzyme 1, and activating transcription factor 6), thereby activating the cell survival unfolded protein response signaling pathways. This response attempts to maintain proteostasis by reducing general protein translation and increasing the production of molecular chaperones. Severe endoplasmic reticulum stress leads to apoptosis of renal epithelial cells. Endoplasmic reticulum stress also increases the expression of T-cell death-associated gene 51 (TDAG51), also known as pleckstrin homology-like domain, family A member 1. This increases the expression of TGF-β (transforming growth factor) receptor 1, which leads to the splicing of the *xbp1* (X box protein) gene. Spliced XBP1 protein leads to the activation of pro-fibrotic genes, and the development of renal interstitial fibrosis [55].

Acute kidney injury superimposed on CKD may accelerate the progression to kidney failure. One of the ways it may do this is by impairing nicotinamide adenine dinucleotide (NAD+) production, by repressing transcription of the gene for quinolinate phosphoribosyl transferase (QPRT, a bottleneck enzyme of de novo NAD+ biosynthesis), as part of the endoplasmic reticulum stress response [56]. A high urinary quinolinate-to-tryptophan ratio can serve as an indirect indicator of impaired QPRT activity and reduced de novo NAD+ biosynthesis in the kidney [56].

#### 2.2.5. Carbamylation

The buildup of urea in CKD may drive the progression of disease via carbamylation [57]. Carbamylation is an irreversible nonenzymatic post-translational modification of proteins from the reaction between isocyanic acid and amino groups on lysine residues or the N-terminal extremity of proteins. Isocyanic acid derives mainly from spontaneous dissociation of urea into ammonia and cyanate (Figure 2). Cyanate is converted into its tautomer isocyanic acid, which is highly reactive and immediately binds to proteins and thus moves the equilibrium toward dissociation. The elevated levels of urea in CKD patients increases cyanate generation and therefore protein carbamylation. Isocyanic acid may also be formed from thiocyanate under the action of myeloperoxidase in the presence of hydrogen peroxide released by leukocytes in sites of inflammation. Carbamylated protein malfunction triggers unfavorable molecular and cellular responses and may accelerate the progression of kidney disease [57].

#### 2.2.6. Ferroptosis

The process of ferroptosis may be important in the progression of CKD [58]. Ferroptosis is a form of regulated cell death driven by iron-dependent phospholipid peroxidation and oxidative stress. With CKD, iron accumulates in renal tubular cells. If lipid peroxide repair capacity by the phospholipid hydroperoxidase, glutathione peroxidase 4 (GPX4), is lost in the presence of this iron, ferroptosis may ensue. If the uptake of cystine via the cystine/glutamate antiporter, system x_c_^−^ is lost, or glutathione synthesis is otherwise impaired in the presence of this iron, ferroptosis may occur. In animal models of CKD, an iron-restricted diet exerts a renal protective effect by inhibiting oxidative stress and aldosterone receptor signaling [58].

#### 2.2.7. DNA Damage and Repair

Insufficient response to DNA damage leads to various insults that enhance apoptosis or result in a dysfunctional phenotype [59]. Several studies have shown a maladaptive response of renal tubular cells during AKI and suggest that an insufficient response to DNA damage could accelerate CKD [10,59,60]. Many studies have described DNA damage as a hallmark of various forms of renal damage characterized by dysfunctional cell cycle proteins of G1/S and G2/M checkpoints and subsequent cell cycle arrest through the activation of p53 or p21 signaling cascades [61,62]. These mutated cells become senescent and have a specific secretome-defined phenotype (senescence-associated secretory phenotype; SASP), which is accompanied by genomic damage and epigenetic abnormalities [63]. The synthesis and release of SASP factors is associated with the activation of the transcription factors, nuclear factor kappa B (NF-κB), and CCAAT enhancer binding protein β (C/EBPβ) [64]. Thus, both altered DNA damage response and NF-κB activation significantly contribute to establishing and maintaining SASP, which produces and releases more senescent secretomes, contributing to functional deterioration of neighboring cells and accelerating the process of cellular death.

#### 2.2.8. Epigenetic Modifications

Several factors such as uremic toxins, oxidative stress, and inflammation increase the prevalence of epigenetic changes and enhance the progression to CKD [65,66]. Inflammation, a major factor in the pathogenesis of CKD, correlates with global DNA methylation [67]. Indeed, a significant association has been identified between DNA methylation and the prevalence and incidence of CKD [66]. Experimental studies observed that hypomethylation in the promoter region of the connective tissue growth factor (*ctgf*) gene is associated with reduced GFR and declined renal function [68]. Data from experimental studies in rats and mice indicate that altered DNA methylation is an important mechanism that initiates the transition from AKI to CKD [69]. In models of unilateral ureteral obstruction (UUO), it was observed that hypermethylation of the Klotho promoter by TGFβ decreased Klotho protein expression, which led to tubular and interstitial fibrosis. Similarly, hypermethylation of the *Vegfa* gene promoter led to reduced VEGF-A signaling, which can lead to capillary collapse and subsequent hypoxia and fibrosis [70]. In ischemic reperfusion rat models, aberrant methylation of the complement C3 promoter region in tubular epithelial cells activated the complement system. This is strongly associated with inflammation and accelerated renal decline [63,71]. Other studies showed that complement component, C5a, promotes DNA hypomethylation of several genes that have integral roles in the initiating cell cycle arrest and senescence [63]. Aberrant hypermethylation of laminin genes has been reported to cause the development of glomerulosclerosis and tubulointerstitial fibrosis in older kidneys, while aberrant methylation of the *rasal1* (Ras protein activator-like 1) gene induced the activation of the Ras–GTPase pathway in fibroblasts, leading to proliferation and fibrosis [72].

In addition to DNA methylation, numerous studies suggest that RNA interference via microRNA (miRNA) is a key factor involved in the pathogenesis and progression of CKD [73]. Several miRNAs, which are involved in post-transcriptional regulation of gene expression, have been linked to inflammation and fibrosis and may enhance the progression of CKD [66,74,75]. One of the most studied miRNAs is miR-192, which plays a role in the expression of profibrotic genes [75]. Upregulated expression of miR-192 leads to the activation of TGF-β and Smad3 signaling pathways, which lead to renal fibrosis through the deposition of collagen and fibronectin [75]. Similarly, the activation of TGF-β pathways increases histone methylation and increased expression of genes involved in deposition of extracellular matrix proteins [75].

#### 2.2.9. Cellular Senescence

Cellular senescence appears to play an important role in the pathogenesis of CKD [76,77]. The accumulation of senescent cells may be responsible for insufficient repair capacity and functional loss. Senescent cells accumulate in the renal parenchyma, leading to tissue deterioration, fibrosis, and aberrant signaling in different types of cell populations [29,69,78]. Senescent cells of renal system express several markers, such as cell cycle arrest proteins of G1/S and G2/M checkpoints such as p16INK4A, p21WAF/CIP1, p27KIP1, and p53, but they do not express proliferation markers such as Ki67 [79,80,81]. Interestingly, selective ablation of senescent (p16INK4a-positive) cells in transgenic mice is linked to diminished expression of TNF-α, IL-6, and IL-1α in many tissues, including the kidney [79]. Cellular senescence has been linked to telomere shortening [82], which upregulates a DNA damage response and activates phosphatidylinositol 3 kinase-like kinases and Rad3-related kinases that lead to p53 activation. Active p53 upregulates transcription of pro-apoptotic genes and/or genes that inhibit cyclin-dependent kinase (i.e., p21^cip1/waf1^). Activation of p21^cip/waf1^ may cause permanent cell cycle arrest [62,80].

Importantly, senescent cells have a specific secretome-defined, senescence-associated secretory phenotype (SASP), which includes a broad range of pro-inflammatory cytokines, chemokines, growth factors, and matrix-degrading factors (e.g., IL-6, IL-1α, IL-1β, chemokine ligand 1 (GROα; CXCL1), connective tissue growth factor (CTGF), plasminogen activator inhibitor 1 (PAI-1), C-C motif chemokine 2 (CCL2)) [79,80,83]. Recent studies report that the expression of integrin β3 increased significantly in senescent cells, which led to the activation of p53 and the secretion of TGF-β [84]. These molecules and others, such as TNF-α, IL-6, and monocyte chemoattractant protein 1 (MCP-1), can promote an inflammatory microenvironment and might be important drivers of inflammation-related injury and enhance progression of CKD [83,85].

#### 2.2.10. Inflammation

Inflammation and fibrosis contribute to the progression of CKD via many pathways in glomeruli, tubules, and interstitium. In the final common pathway to end-stage renal disease, nephron loss causes hyperperfusion and high glomerular capillary hydrostatic pressure in remaining nephrons. This results in injury to the major cell types within glomeruli: endothelial cells, podocytes, and mesangial cells [86]. Injured endothelial cells detach from the basement membrane, express more leukocyte adhesion molecules, and secrete more proinflammatory cytokines. Injured podocytes also detach from the basement membrane, allowing proteinuria, associated with increased angiotensin II, aldosterone, and TGF-β. Injured mesangial cells proliferate and synthesize extracellular matrix constituents, MCP-1, CGTF, and TGF-β. Monocytes recruited to injured glomeruli become macrophages, and glomerular inflammation leads to glomerulosclerosis [87].

Interstitial inflammation and fibrosis also contribute to the progression of CKD. Lymphocytes and macrophages infiltrate the renal interstitium. Lymphocytes are recruited into the interstitium early. Monocytes are recruited into the interstitium, where they become macrophages. Macrophage infiltration after nephron loss is chiefly in tubulointerstitial regions. The CC chemokine receptor type 1 is important in interstitial but not glomerular recruitment of leukocytes. Dendritic cells appear in the interstitium, with peak concentration at one week after nephron loss. Mast cells are identifiable in areas of tubulointerstitial inflammation and fibrosis [88]. Myofibroblasts secrete the components of extracellular matrix, leading to fibrosis. The predominant source of these myofibroblasts is from pericytes around blood vessels and resident fibroblasts, with a minor contribution from de-differentiated proximal tubule cells [89].

In the progression of chronic kidney disease, the cell type probably most often central to the various processes involved is the macrophage [90]. The mononuclear cell chemokine, CCL2, mediates migration of monocytes to the injured kidney. CCL2 blockade attenuates glomerular and interstitial infiltration of pro-inflammatory macrophages, but other chemokines such as CX3CL1, CXCL16, and macrophage migration inhibitory factor (MIF), also contribute to macrophage recruitment in kidney disease [85]. Opposing this recruitment, the mononuclear cell chemokine C-C motif chemokine 5 (CCL5) constrains CCL2 expression, macrophage infiltration, and kidney damage and fibrosis in hypertension via blood pressure-independent mechanisms. This balance illustrates the complex network of overlapping chemokines working to maintain renal health [85].

Macrophages polarized to the M1 phenotype play a pathogenic role in inflammatory renal injury, and macrophages polarized to the M2 phenotype play a pathogenic role in the follow-on renal fibrosis. Recruited macrophages produce a range of cytokines, including TNF-α and interferon-γ (IFN-γ), which increase M1 polarization and the progression of CKD. Damage-associated molecular patterns (DAMPs) from renal parenchyma such as high-mobility group protein B1 (HMGB1) and C-reactive protein (CRP) also augment the renal accumulation of pro-inflammatory macrophages. Additionally, DAMPS released from damaged renal tubular epithelial cells (RTEC) have been shown to increase the expression of pattern recognition receptors (PRR) on healthy RTEC. This leads to the expression of pro-inflammatory cytokines, which leads to further recruitment of inflammatory monocytes and macrophages [91]. There is experimental evidence that macrophage polarization to a pro-inflammatory phenotype, associated in turn with increased pro-inflammatory cytokines and renal inflammation, is promoted by a high salt intake, suggesting that this is one of the mechanisms by which a low-salt diet ameliorates the progression of CKD [92].

Macrophage phenotype is partly determined by prostaglandins in the microenvironment. Arachidonic acid metabolism into prostaglandins plays a role in renal inflammation and fibrosis. The lipoxygenase family of enzymes convert polyunsaturated fatty acids into bioactive lipid eicosanoids such as hydroxyeicosatetraenoic acids, hydroxyeicosaoctadecaenoic acids, leukotrienes, lipoxins, and resolvins [93]. The enzyme 15-lipoxygenase worsens inflammation and fibrosis in a rodent model of chronic kidney disease. Silencing 15-lipoxygenase promotes an increase in M2c-like wound-healing macrophages in the kidney and alters kidney metabolism, protecting against anaerobic glycolysis after injury [93].

The presence of free fatty acids may drive some of the inflammation causing the progression of chronic kidney disease. Renal tubular injury from free fatty acids may be partly mediated by increased expression of the CD36 scavenger receptor due to increased expression of the peroxisome proliferator-activated receptor (PPAR)-γ nuclear transcription factor [94]. Renal tubular injury and inflammation due to increased circulating free fatty acids could partly explain the accelerated progression of CKD in patients with insulin resistance (type 2 diabetes mellitus) and obesity. The decelerated progression of CKD by sodium-glucose co-transporter 2 (SGLT2) channel inhibitors may be partly from attenuated CD36 expression from downregulated PPAR-γ [94]. IL-33 may be an important driver of progressive CKD [95]. IL-33 is a member of the IL-1 cytokine family and exerts pro-inflammatory and pro-fibrotic effects via the suppression of tumorigenicity 2 (ST2) receptor, which, in turn, activates other inflammatory pathways. Recent studies have shown that a sustained activation of the IL-33/ST2 pathway promotes the development of renal fibrosis [95]. This pathway is a potential target for therapeutic intervention.

Excess amino acids from the heavy nutritional load of a high-protein diet are thought to increase renal inflammation and fibrosis primarily indirectly by increasing glomerular hyperfiltration, but also partly by increasing proinflammatory gene expression. Later in the course of chronic kidney disease, excess amino acids promote fibrosis by increasing TGF-β. Whatever the mechanisms, a low-protein diet slows the progression of chronic kidney disease [96].

#### 2.2.11. Fibrosis

Renal fibrosis is the end result of nearly all progressive renal diseases. Fibrosis is a maladaptive repair process associated with chronic inflammation. It is characterized by progressive remodeling and destruction of renal tissue in an attempt to replace injured cells. Although the initial stages of this repair process may be beneficial, prolonged activation of growth factors and cytokines leads to the replacement of normal renal parenchyma with collagen and other connective tissue fibers [97].

It is generally accepted that fibroblasts and myofibroblasts are key cells involved in renal fibrosis. Under normal conditions, fibroblasts synthesize many of the constituents of the extracellular matrix. It is well accepted that myofibroblasts are activated fibroblasts. Renal injury leads to numerous stimuli that may cause transformation of fibroblasts to myofibroblasts. These stimuli include the production of inflammatory cytokines (e.g., TGF-β, platelet-derived growth factor (PDGF), fibroblast growth factor 2 (FGF-2)), hypoxia, and cell contact with leukocytes and macrophages. Interestingly, renal fibroblasts maintain their activated phenotype in a setting of fibrosis even if the initial cause is no longer present [98].

Published evidence suggests that renal epithelial cells play an important role in renal fibrosis due to epithelial-to-mesenchymal transition [98,99,100]. This transition appears to be induced by interleukin-like epithelial mesenchymal transition inducer (ILEI) in response to TGF-β1 through Akt (protein kinase B) and ERK (extracellular signal-regulated kinase) pathways [101]. Following this transition, renal tubular epithelial cells lose their normal morphology, tight junctions, and epithelial cell markers (e.g., E-cadherin), and begin expressing mesenchymal markers such as α-smooth muscle actin (α-SMA) and vimentin. These alterations facilitate the progression of renal interstitial fibrosis and CKD [101].

PDGFs play an important role in the processes that lead to renal fibrosis [102]. PDGF receptor-β (PDGFR- β) is a tyrosine-kinase receptor for PDGF-B and PDGF-D. Upon activation, PDGFR-β induces downstream signaling that triggers cell proliferation, migration, and differentiation, leading to extracellular matrix deposition. There is experimental evidence that PDGFR-β activation alone is sufficient to induce progressive renal fibrosis and renal failure, key aspects of CKD [103]. PDGFR-β is a potential target for therapeutic intervention to slow the progression of kidney disease.

## 3. Molecular Effects of Environmental Toxicants on CKD Progression

As CKD progresses and GFR decreases, the ability of patients to eliminate metabolic wastes, xenobiotics, and toxicants declines significantly. Considering that the current environment is contaminated with numerous toxicants, it is important to understand how patients with a reduced ability to excrete toxicants are affected by environmental nephrotoxicants. Of particular concern is exposure to nephrotoxic heavy metals that are present throughout the environment. Today’s environment is heavily contaminated by toxic metals such as arsenic, cadmium, and mercury; therefore, human exposure to one or more of these toxicants is nearly unavoidable. Indeed, the World Health Organization included these three metals on the list of top ten chemicals of public health concern [104]. A thorough understanding of the way in which these metals are handled by diseased kidneys is necessary to manage this important global health problem.

### 3.1. Arsenic

Arsenic (As) is a naturally occurring metalloid found in the Earth’s crust [105]. As may exist in inorganic (arsenopyrite, pentavalent arsenate, trivalent arsenite) and organic (e.g., monomethylarsonic acid, dimethylarsonic acid, trimethylarsonic acid, arsenobetaine) forms [106]. Inorganic As seeps into groundwater reservoirs via environmental weathering of ores and contaminates underground aquifers. Inorganic As, in the form of arsenite (As(OH)_3_) and arsenate (H_3_AsO_4_), is often found in water wells located in rocky terrain around the world [107,108]. Anthropogenic activity has also contributed to increased arsenic pollution through use of pesticides, handling of arsenate-containing wood preservatives, and semiconductor manufacturing [109]. While it can be inhaled or absorbed via dermal contact, human exposure occurs primarily via drinking water contaminated with inorganic arsenic [110].

Exposure to As is a serious global health concern that is associated with numerous health effects. Acute physiological effects of As exposure include various multiorgan symptoms ranging from colicky abdominal pain to encephalopathy [111]. Chronic exposure to arsenic is known to cause bladder, skin, and lung cancer. It is linked with kidney, liver, and prostate cancer and may cause numerous other health effects [112,113]. A meta-analysis of literature related to heavy metal exposure and CKD reported a link between exposure to As and risk of proteinuria, an early sign of CKD [114]. Similarly, it has been reported that exposure to heavy metals such as As may increase the risk of developing CKD [114]. Indeed, chronic exposure to As has been shown to result in glomerulonephritis, acute tubular necrosis, albuminuria, and renal papillae necrosis [115,116].

Exposure to As can lead to significant oxidative stress, which can exacerbate renal injury and enhance the progression of CKD (Figure 3). Studies in rats have shown that As increases production of ROS, which enhances the expression of inflammatory cytokines through the NF-κB pathway [117]. This induces apoptosis primarily by decreasing Bcl-2 and Bcl-xl (Bcl-2 associated protein) expression while concomitantly increasing expression of p53 and Bax in As-treated rats [117]. Recent studies in mice showed that exposure to As activates the MAPK/NF-κB and NRF2 pathways in kidney. While activation of these pathways may improve cell survival, exposure to As also led to increased activity of myeloperoxidase and increased expression of inflammatory cytokines, such as IL-1α, IL-6, IL-12, and TNF-α, which may lead to an inflammatory response [118,119]. Furthermore, elevated expression of inflammatory cytokines was shown to disrupt homeostasis of helper T cell populations (Th1/Th2/Th17/Treg). The balance among T cells populations is critical to maintain proper immune function. Exposure to As alters the balance and leads to inflammation and immunosuppression [118,120]. Use of newer immunotherapeutics and their impact on kidney function is a new and expanding area of research. Some immunotherapeutics have been reported to exacerbate or cause renal damage [121]. The complexity of renal function and inflammation makes this an interesting new area of research with great potential.

Studies using cultured myoblasts indicate that As exposure leads to apoptosis through pathways involving ROS, mitochondrial dysfunction, and endoplasmic reticulum stress [122]. A similar pathway may play a role in As-induced nephrotoxicity. Interestingly, trivalent forms of As have been shown to inhibit the production of glutathione, which may lead to unrestricted oxidative stress [123]. Arsenic can cause lipid peroxidation and damage mitochondrial membranes, leading to the formation of peroxyl radicals and dimethylarsenic radicals, and eventual cell death [123].

### 3.2. Cadmium

Cadmium (Cd) naturally exists in zinc, lead, and copper ores as a divalent cation in the Earth’s crust and marine environments at low concentrations and accumulates in air water and soil through volcanic activity and erosion [124]. While trace amounts of Cd in the environment are byproducts of these processes, the majority of environmental Cd is the result of industrial and agricultural use [124]. Soluble Cd ions from phosphate fertilizers can contaminate water and soil and subsequently accumulate in aquatic organisms or plants such as tobacco, grains, and root vegetables [125,126]. Because of accumulation in tobacco plants, individuals who smoke are exposed to significant levels of Cd through the inhalation of cigarette smoke. Cigarettes may contain 1.8–2.5 µg/g Cd [127], and data from the National Health and Nutrition Examination Study (NHANES) reported that smokers had average blood and urine Cd levels of 0.376 µg/L and 0.232 µg/L, respectively [128].

Following exposure, Cd is absorbed readily by epithelial cells in the gastrointestinal tract and lungs. Once ingested, ionized Cd^2+^ binds to albumin and the resulting complexes are then transported to target organs, including the kidney, bone, liver, and lung. Cd is taken up into hepatocytes via Ca^2+^ channels and membrane transporters [129]. Within the cell, Cd has been shown to impair electron transport chain complexes II and III, which impedes electron flow and generates ROS [112]. The generation of ROS promotes binding of the metal-regulated transcription factor 1 (MTF-1) to metal response elements (MRE), which subsequently activates transcription of metallothionein (MT) [110]. Intracellularly, MT binds to Cd to create a MT–Cd complex, and a fraction of the complex is exported into the circulation [130]. MT–Cd complexes are filtered freely by the glomerulus and are reabsorbed by proximal tubular epithelial cells via multiple mechanisms, including megalin and cubilin receptor-mediated endocytosis, ZIP8 and ZIP14, and the divalent metal transporter 1 (DMT1) [131,132].

Environmental and occupational exposure to low levels of Cd has been shown to cause renal tubular injury [133]. Owing to its toxic renal effects, chronic exposure to Cd increases the risk of developing CKD from 10% in the average population to 25% in exposed individuals [134]. Analyses of NHANES data showed that individuals with blood Cd levels over one mcg/L had a significantly higher association with CKD and albuminuria [135]. It is clear that exposure to Cd reduces GFR and impairs overall renal function. Here, we summarize the molecular mechanisms that underlie these pathologies and contribute to the development and/or progression of CKD (Figure 4).

As discussed in the previous sections, diabetes is a major risk factor for developing CKD. Exposure to Cd appears to increase the risk of developing diabetes in some individuals [136]. Diabetic mice (db/db) exposed to Cd were found to be hyperglycemic and exhibited an increase in white adipose tissue and weight gain [137]. Interestingly, exposure of these mice to Cd also decreased serum leptin levels, which may enhance appetite and lead to weight gain. In contrast to these findings, a recent study using streptozotocin-induced diabetic mice (C57BL/6) exposed to Cd showed that body weight decreased after exposure [138]. This variation could be due to differences in the frequency and dose of Cd exposure. This study, however, confirmed findings that exposure of hyperglycemic animals to Cd enhances the risk of renal injury [138]. Specifically, in vitro studies using cultured podocytes indicate that exposure to Cd under hyperglycemic conditions leads to mitochondrial dysfunction and ROS. Podocyte viability is reduced, leading to apoptosis, fibrosis, and decreased renal function [138]. These studies suggest that exposure of diabetic individuals to Cd may exacerbate renal injury and lead to CKD or enhance the progression of CKD by causing additional injury.

Hypertension is another major risk factor for the development of CKD. A meta-analysis study of published literature found a positive association between hypertension and Cd levels in blood and hair [139]. Indeed, Cd has been shown to decrease plasma levels of atrial natriuretic peptide (ANP) [140], an important regulator of blood pressure. Cd appears to reduce the affinity of the ANP receptor for ANP and also decreased the number of binding sites available [141]. Reduced levels of ANP and reduced sensitivity of the receptor may decrease the ability to regulate blood pressure, which may lead to hypertension and subsequent renal injury. The development of hypertension is characterized by low ANP plasma concentrations [142] and its suppressed ability to regulate blood pressure via inhibiting the renin–angiotensin–aldosterone system. In addition to causing vasodilation of the afferent arterioles, ANP binds to natriuretic peptide receptor-A, catalyzing the conversion of GTP to cGMP. cGMP phosphorylates and allosterically binds to basolateral sodium-potassium ATPase channels and apical cyclic nucleotide-gated, heterometric channels of transient receptor potential V4 and P2 [143]. While the mechanism remains entirely unclear, damage to the kidney’s response to ANP may potentially be mediated by Cd-induced oxidative damage. In one study, increased production of thiobarbituric acid reactive substance after exposure to Cd was accompanied by decreased glomerular filtration rate and increased creatinine levels; upon bolus injection of ANP in compromised rats, high blood pressure and low glomerular filtration rate remained remarkably uncorrected. Without the counteraction of salt and fluid retention from ANP due to compromised receptor response, Cd may play a role in exacerbating hypertensive conditions precipitating kidney injury and eventual CKD [143].

In addition to the association with diabetes and hypertension, exposure to Cd has been shown to cause generalized cellular injury in renal epithelial cells. A major consequence of Cd exposure is intracellular oxidative stress. Studies using male Sprague Dawley rats found swelling, deformation, and vacuolation in mitochondria of renal tubular epithelial cells. In addition, expression of superoxide dismutase 2 (SOD2), found in mitochondria, decreased, indicating an inability to counteract the production of ROS. Indeed, increased cellular content of ROS was accompanied by increased expression of cytoplasmic superoxide dismutase 1 (SOD1). Interestingly, expression of catalase was reduced, which would prevent cells from responding appropriately to oxidative stress [144]. Other studies reveal the association of Cd exposure to substantial activity reduction in antioxidant enzymes, including superoxide dismutase, catalase, and glutathione reductase, that may amplify the progression of chronic kidney disease from oxidative species overwhelming antioxidants [145].

In addition to oxidative stress, Cd has been shown to induce ER stress and autophagy (via BNIP3) in HK2 cells and SD rats [146]. Studies in cultured rat pheochromocytoma cells (PC-12) showed that exposure to Cd enhanced autophagy [147]. In contrast, studies in mice exposed to Cd showed that protein components of autophagosomes (e.g., p62, Sirt6, and LC3-II) accumulated in the cytoplasm of renal tubular cells rather than participate in the formation of autophagosomes. This resulted in the inhibition of autophagy and the initiation of apoptosis [148]. Similarly, a study in cultured proximal tubular cells reported that treatment with Cd led to the accumulation of p62 in the cytoplasm and the inhibition of autophagy. Furthermore, it was reported that elevated levels of p62 led to increased nuclear translocation of Nrf2 [149]. Persistent activation of Nrf2 can lead to lysosomal dysfunction, which prevents fusion of lysosomes and autophagosomes [150]. Cd may also induce apoptosis in renal epithelial cells through p-53 mediated, DNA damage autophagy modulator (DRAM) and BAX signaling [151]. Cd has also been shown to activate inflammatory cytokines, such as NF-κb, TNF-α, and iNOS, and induce necroptosis [152,153].

Exposure to Cd has also been shown to disrupt cadherin-dependent cell adhesion in proximal tubular cells. Alterations in cellular adhesion have been shown to alter the membrane localization of the Na^+^K^+^-ATPase, which can lead to alterations in transport [154]. Similarly, other studies in proximal tubules showed that exposure to Cd decreased expression of SGLT1 and SGLT2. This decrease was attributed to the replacement of Zn in the Sp1 DNA binding domain, which reduced activation of SGLT1 and SGLT2 promoters [155].

Cd is a toxic metal to which humans continue to be exposed throughout their lives. Continued studies related to Cd-induced cellular injury in renal tubular epithelial cells are necessary to understand how exposure to this metal affects patients with renal disease.

### 3.3. Mercury

Mercury (Hg) is a toxic metal found in various environmental and occupational settings. It may exist in an elemental (metallic), inorganic, and/or organic form. Elemental mercury (Hg^0^) is particularly unique because it exists as a liquid at room temperature. Inorganic mercury is usually found as mercurous (Hg^1+^) or mercuric (Hg^2+^) ions salts. Organic forms of mercury include phenylmercury, dimethylmercury, and monomethylmercury (MeHg), which is the most common form encountered by humans. The majority of human exposure is due to the ingestion of contaminated food. Upon ingestion, MeHg is absorbed readily by enterocytes along the gastrointestinal tract [1], after which they can enter systemic circulation and be delivered to target organs. Within biological systems, a fraction of MeHg is slowly transformed to Hg^2+^ [156,157,158,159].

Exposure to all forms of Hg has been shown to have significant renal effects (Figure 5). Experimental models (uninephrectomy) of early-stage CKD suggest that acute renal injury is more pronounced in uninephrectomized rats exposed to a nephrotoxic dose of HgCl_2_ than in corresponding sham rats [160,161,162]. It was found that mercury-induced proximal tubular necrosis was more extensive in 50% nephrectomized animals than in sham animals. Additionally, the urinary excretion of cellular enzymes and plasma proteins, including lactate dehydrogenase, γ-glutamyltransferase, and albumin, was greater in uninephrectomized animals than in sham animals [162,163]. Interestingly, when 75% nephrectomized rats were used as models of late-stage CKD, it was found that the accumulation of mercury per g kidney is significantly greater in 75% nephrectomized rats than in sham rats, suggesting that cellular accumulation of Hg may be greater in the remnant renal mass from 75% nephrectomized animals than in kidneys of sham animals [164].

In humans, chronic exposure to mercury has been associated with glomerulonephritis, particularly membranous nephropathy [165,166]. Membranous nephropathy is characterized by tissue damage due to activation of membrane attack complexes (MAC) by antigen–antibody complexes deposited on the glomerular basement membrane (GBM) [167]. This damage results in podocyte damage and disruption of the anionic charge barrier, leading to massive proteinuria [167]. Analyses of patient biopsies found that patients with Hg-induced membranous nephropathy exhibited more mesangial deposits and smaller podocyte foot processes than patients with idiopathic membranous nephropathy. Interestingly, podocyte effacement was less severe in Hg-induced cases than in idiopathic cases [168]. In vitro studies have shown that exposure to Hg leads to autoimmune disease characterized by anti-GBM antibodies, glomerular deposits of immunoglobin G (IgG), proteinuria, and acute tubulointerstitial nephritis [169,170,171]. Studies in Brown Norway rats have shown that exposure to HgCl_2_ leads to a T-cell dependent autoimmune syndrome that leads to the production of anti-laminin antibodies that interact with the GBM [172,173]. Exposure to HgCl_2_ leads to the appearance of non-antigen-specific CD8+ T-cells [174]. Additional studies in Brown Norway rats showed that RT6+ T cells decreased, which inversely corresponded with the autoimmune response to the GBM [175]. Escudero et al. showed that the HUTS-21 epitope of the beta-1 integrin on lymphocytes appears to be involved in Hg-induced nephritis by promoting lymphocyte infiltration into renal interstitium and deposition of anti-GBM antibodies [176].

Exposure to Hg may also play a role in the development of hypertension. A study of non-Hispanic Asians using NHANES data found that higher blood Hg levels were associated with hypertension [177]. Studies using spontaneously hypertensive rats (SHR) found that exposure to Hg accelerated the development of hypertension by increasing the production of nitric oxide and other ROS [178,179]. However, it appears that Hg also induces vasoprotective mechanisms such as increased plasma levels of nitric oxide and hydrogen peroxide to counteract other vasoconstrictive effects [179]. In addition, plasma levels of angiotensin-converting enzyme (ACE) were found to be increased in SHR rats following exposure to Hg [180], which can lead to vasoconstriction and hypertension. In a recent review, Habeeb et al. outlined the molecular mechanisms by which Hg exposure leads to hypertension [181]. Hg has been shown to increase atherosclerosis as well as stimulate the proliferation of vascular smooth muscle cells, which would further increase the risk of hypertension.

The effects of Hg on renal tubule epithelial cells can be detrimental to total renal function. Studies in cultured proximal tubular cells have demonstrated that exposure to Hg induces significant cellular alterations [182,183]. Specifically, the most profound modifications were noted as increased oxidative stress, cytoskeletal rearrangements, increased intracellular calcium, and reduced cellular viability.

In the mitochondria, mercury-induced oxidative stress has been shown to disrupt the overall structure, leading to swelling, destruction of mitochondrial membrane potential, altered membrane, and increased release of Cytochrome C [184]. Exposure of human embryonic kidney epithelial (HEK-293T) cells to HgCl_2_ revealed a decrease in cell viability due to a downregulation in the expression of the silent information regulator (Sirt1) and PGC-1α signaling pathway, a key mechanism in mitochondrial homeostasis [185].

Cytoskeletal alterations have been detected following exposure of normal rat kidney cells (NRK-52E) to MeHg [186]. These alterations are a result of epigenetic modulation of matrix metalloproteinase 9 (MMP9) via demethylation of its regulatory site. The subsequent increased expression of MMP9 led to loss of cell-to-cell adhesion and disturbances in cytoskeletal proteins such as F-actin, vimentin, and fibronectin [186]. Similarly, exposure of NRK-52E cells to HgCl_2_ also led to loss of cytoskeleton integrity [182].

The ER is another cellular target in acute HgCl_2_ toxicity. Experiments in NRK-52E cells showed that ER stress, as indicated by expression of GRP78 (78-kDa glucose regulated protein) and CHOP (C/EBP homologous protein), is a marker of renal cell injury [187]. GRP78 is an ER chaperone, which is upregulated upon stress; however, if the ER experiences prolonged stress, CHOP, a transcription factor specializing in regulation of apoptosis-related genes, will also be upregulated. Both of these proteins are positively correlated with renal damage. In addition, HgCl_2_ has been shown to enhance the activity of Caspase 3 and the expression of IRE1_a_ (inositol-requiring enzyme 1), GADD-153 (growth arrested and DNA damage-inducible gene 153), and Caspase 12, resulting in the death of tubular and glomerular cells [188].

Exposure to Hg also affects the activity of various transporters, which may lead to tubular injury and renal disease. Studies in Wistar rats exposed to a low dose of HgCl_2_ showed that Hg inhibited the Na^+^/H^+^ exchanger (NHE3) in proximal tubular cells. It was suggested that Hg enhanced phosphorylation of NHE3 and thereby reduced its activity [189]. NHE3 is the main isoform of the Na^+^/H^+^ exchanger in the proximal tubule and it plays a major role in the reabsorption of sodium from the lumen. Alterations in the activity of NHE3 could indirectly affect reabsorption and secretion of important molecules and fluid. In addition, HgCl_2_ has been shown to inhibit Na-K-ATPase [190], which would alter solute gradients necessary for water reabsorption and lead to increased urinary output, a common sign of renal injury. Mercury has also been shown to bind to cysteine residues in aquaporin 1 (AQP1) [191], located in the proximal tubule and thin limbs in the loop of Henle [192], and inhibit its activity. AQP1 facilitates reabsorption of 70% of water from the ultrafiltrate entering the proximal tubule [193]. Therefore, the inhibition of AQP1 is also a likely cause of increased urinary output following Hg intoxication.

Collectively, the results of these studies suggest that kidneys of animals with reduced renal mass are more susceptible to the toxic effects of Hg. Similarly, individuals who have reduced renal function, due to CKD or other disease processes, may be more susceptible to renal injury following exposure to a nephrotoxicant such as Hg.

## 4. Summary

The pathogenesis and progression of CKD result from numerous physical and molecular changes that create a complex intracellular environment. Physical changes due to hypertrophy and hydrostatic forces may cause injury to cells and lead to an inability of the cells to manage small intracellular changes. Increased oxidative stress, ER stress, DNA modifications, mitochondrial dysfunction, and many other cellular and molecular changes lead to dysregulation of intracellular processes, cellular injury, and eventual cell death. Exposure to environmental toxicants such as heavy metals may lead to additional cellular injury and enhance the progression of CKD. Because of the complexity of the CKD, eliminating one path of pathogenesis may enhance pathogenesis via a different route. Stopping the progression of CKD will likely require a combination of multiple therapies, but each component of combination therapy will likely cause its own negative effects. The only reasonable and attainable goal may be slowing down the progression of this disease.

## Figures and Tables

**Figure 1 ijms-23-11105-f001:**
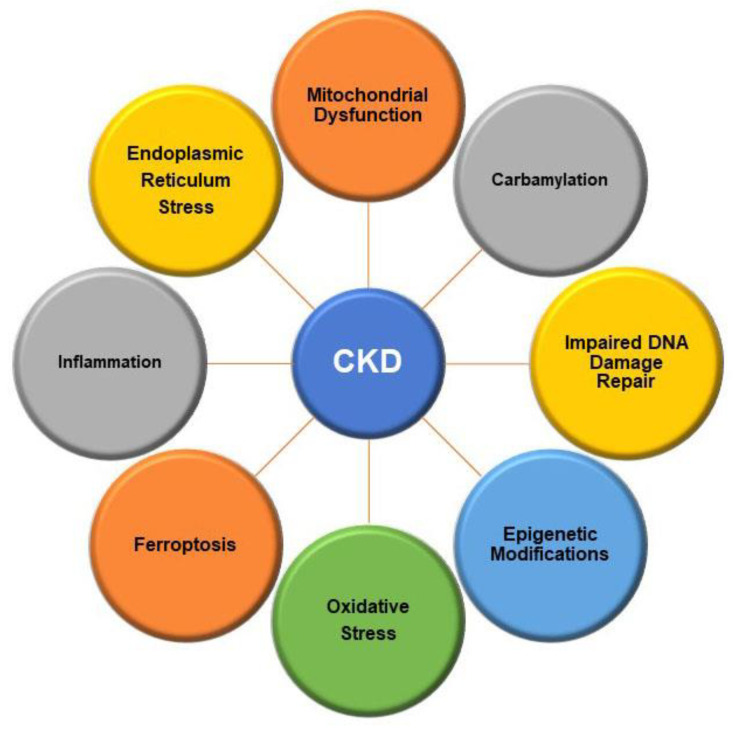
Molecular mechanisms involved in pathogenesis and progression of chronic kidney disease (CKD). CKD is a complex disease that involves dysregulation of multiple physiological processes. This diagram is meant to show the major molecular mechanisms that promote pathogenesis of CKD with the understanding that there are many other mechanisms that may also be involved in this process.

**Figure 2 ijms-23-11105-f002:**
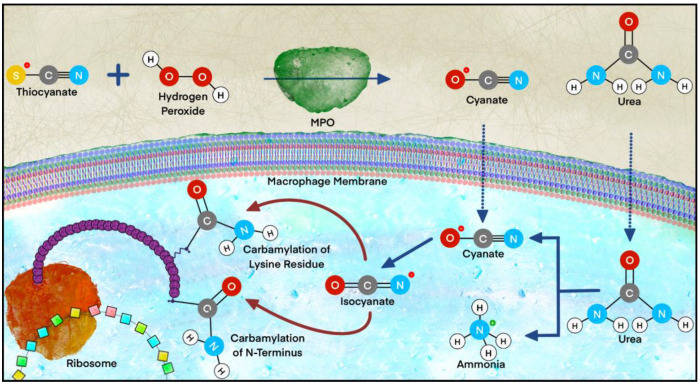
Process of carbamylation. Carbamylation occurs through two primary pathways that converge due to the spontaneous reactivity of isocyanate with lysine residues and the N-termini of nascent polypeptides. The first and predominant pathway is the spontaneous dissociation of urea to cyanate and ammonia. The second pathway is the conversion of thiocyanate and hydrogen peroxide to cyanate under the action of myeloperoxidase. Once cyanate is formed, it is converted into isocyanate, and the spontaneous and irreversible process of carbamylation commences. MPO, myeloperoxidase.

**Figure 3 ijms-23-11105-f003:**
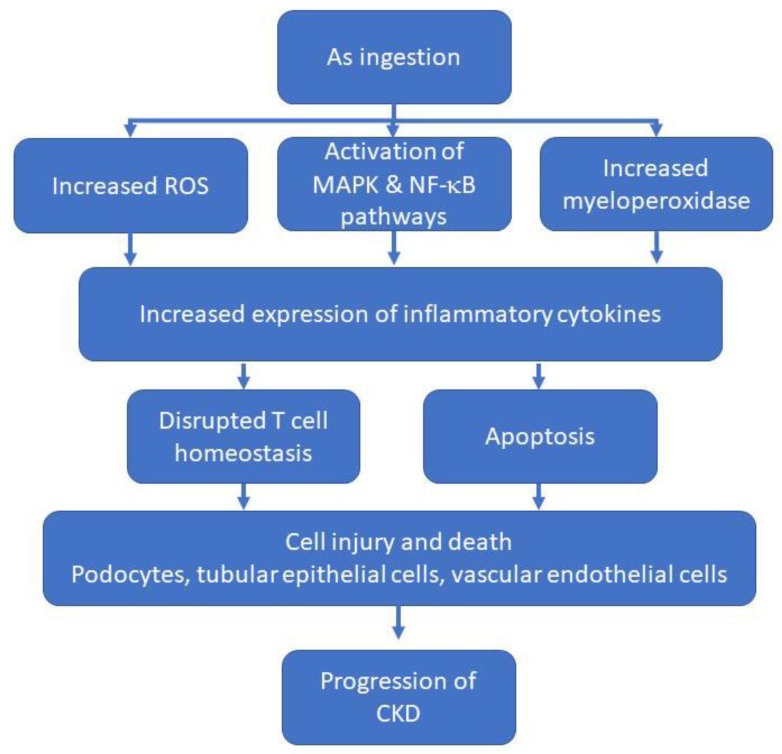
Flowchart summarizing major effects of arsenic (As) exposure in relation to the progression of CKD. As-induced injury is a complex, multifactorial process that cannot be summarized completely in a single figure. Thus, while this figure includes major pathways of As-induced injury, it does not cover all mechanisms and routes of injury.

**Figure 4 ijms-23-11105-f004:**
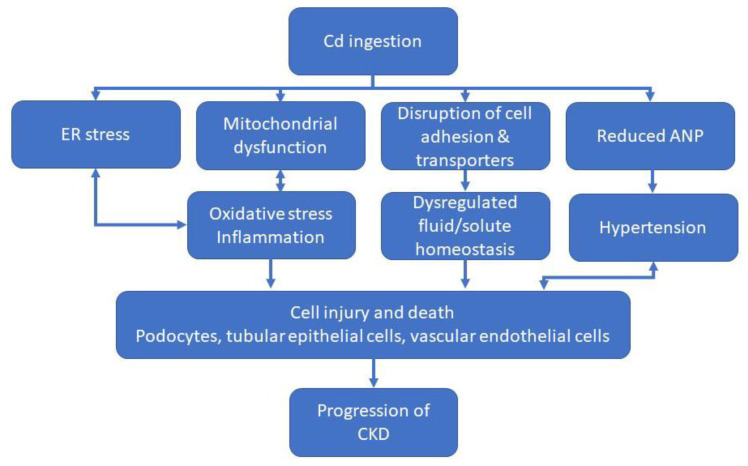
Schematic to summarize major factors that contribute to progression of CKD following exposure to cadmium (Cd). Additional factors not included here also contribute to Cd-induced progression of CKD.

**Figure 5 ijms-23-11105-f005:**
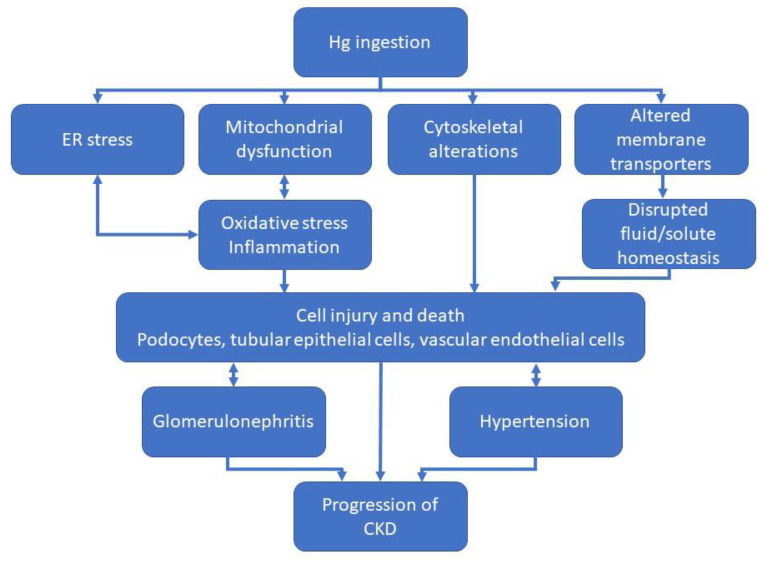
Flowchart outlining major mechanisms involved in mercury (Hg)-induced progression of CKD. Other factors not specified here also play a role in the progression of CKD induced by exposure to Hg.

## Data Availability

Not applicable.

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
