# Peer review of "Molecular Mechanisms of Cellular Injury and Role of Toxic Heavy Metals in Chronic Kidney Disease"

_ijms, 2022, doi:10.3390/ijms231911105_

Round 1

Reviewer 1 Report

It is quite a well-organized review and researchers focused on metal toxicity would be interested in the topic. However, the manuscript should be improved according to the following comments:

- some additional figures would improve the text,

- summarizing tables would help the manuscript as well,

- my major comment relates to the style of the chapters: the authors summarized the knowledge but they did not add discussion / expert opinion into the text, I recommend adding such text to the proper sites of the manuscript. 

Author Response

It is quite a well-organized review and researchers focused on metal toxicity would be interested in the topic. However, the manuscript should be improved according to the following comments:

- some additional figures would improve the text,

Thank you for your suggestion.  We have added three additional figures outlining the effects of each metal on the progression of CKD.

- summarizing tables would help the manuscript as well,

Thank you for your comment. We considered adding tables that showed mediators of CKD but we found that the tables would be quite extensive and cumbersome.  We believe that the material is covered more succinctly within the text.   

- my major comment relates to the style of the chapters: the authors summarized the knowledge but they did not add discussion / expert opinion into the text, I recommend adding such text to the proper sites of the manuscript. 

Thank you for your suggestion. Our goal in writing this review was to summarize the current literature related to the mechanisms involved in the pathogenesis of CKD.  In some areas of the manuscript, we included summary sentences where we try to “connect the dots” between the published data.  We tried to refrain from inserting our opinions throughout the manuscript because we do not want readers to confuse our opinions with published literature. 

Reviewer 2 Report

This is a review on the pathogenesis of and molecular mechanisms of cellular injury in CKD with an additional segment summarizing the role of heavy metals in CKD.  The review is comprehensive with a lot of data.  It could be informative, but I thought that it could be improved as follows:

1. The title is long and complicated. I would suggest: "Molecular mechanisms of cellular injury and role of heavy metals in CKD"

2. Fig. 1: the category "nephrotoxic xenobiotics" is a misfit here, given that those xenobiotics lead to CKD through the other mechanisms listed. It is not a mechanisms, more a cause.

3. Given the amount of literature cited here, I would suggest to have each section started with a summary sentence and concluded with a conclusding paragraph.

4. Line 115: mitochondrial abnormalities due to oxidative stress or oxidative stress due to mitochondrial abnormalities?

5. The role of VEGF in the disease mechanisms is not fully reviewed here. Consider adding all the literature associated with VEGF and glomerular changes.

6. What about the genetic component of CKD, as well as the impact of fetal development?

7. There are many places where the Greek symbols have been lost after IFN, TNF or TGF.

8. The section of Molecular Interventions for CKD is really small and non-informative. It should be either removed or expanded.

9. What about discussing the new generation of biomarkers to detect earlier CKD with the hope to slow progression more efficiently?

10. Summary does not even cover the heavy metal section.

11.  A figure summarizing the site of action of the heavy metals covered in the review would be an excellent addition.

Author Response

This is a review on the pathogenesis of and molecular mechanisms of cellular injury in CKD with an additional segment summarizing the role of heavy metals in CKD.  The review is comprehensive with a lot of data.  It could be informative, but I thought that it could be improved as follows:

  1. The title is long and complicated. I would suggest: "Molecular mechanisms of cellular injury and role of heavy metals in CKD"

Thank you for your suggestion.  We have shortened the title as recommended.

  1. Fig. 1: the category "nephrotoxic xenobiotics" is a misfit here, given that those xenobiotics lead to CKD through the other mechanisms listed. It is not a mechanisms, more a cause.

Thank you for pointing this out. We have edited the figure based on the reviewer’s comment.

  1. Given the amount of literature cited here, I would suggest to have each section started with a summary sentence and concluded with a conclusding paragraph.

We understand the value in the reviewer’s suggestion; however, we believe that the section headings provide a good indication of the subject of the paragraph. Also, we feel that the addition of a concluding paragraph to each section would be cumbersome and lengthen the manuscript considerably.

  1. Line 115: mitochondrial abnormalities due to oxidative stress or oxidative stress due to mitochondrial abnormalities?

We agree that mitochondrial abnormalities can lead to oxidative stress and vice versa. We have edited this section of the text for clarity.

  1. The role of VEGF in the disease mechanisms is not fully reviewed here. Consider adding all the literature associated with VEGF and glomerular changes.

Thank you for noting this omission.  We have included additional information on the role of VEGF in various sections throughout the manuscript.

  1. What about the genetic component of CKD, as well as the impact of fetal development?

Thank you for pointing out this important point. We recognize that there could be a genetic basis for the development of CKD, but we feel that those topics are outside the scope of this review. Topics such as polycystic kidney disease and CKD would be ideal for a future review.

  1. There are many places where the Greek symbols have been lost after IFN, TNF or TGF.

Thank you for noting this.  We believe that the symbols were lost during the file upload.  We have corrected them and we hope they are visible in the current version.

  1. The section of Molecular Interventions for CKD is really small and non-informative. It should be either removed or expanded.

Thank you for your comment. As suggested, we have removed this section.

  1. What about discussing the new generation of biomarkers to detect earlier CKD with the hope to slow progression more efficiently?

We agree with the reviewer that biomarkers are extremely important and relevant. However, several recent comprehensive reviews (Aug 2022, Sept 2022) have been published describing biomarkers for the detection of CKD. Therefore, we feel that the addition of a biomarkers section to the current review would not add to the overall body of literature.

  1. Summary does not even cover the heavy metal section.

Thank you for noting this omission. We have edited the summary section to include heavy metals.

  1. A figure summarizing the site of action of the heavy metals covered in the review would be an excellent addition.

Thank you for this suggestion.  We have added a flowchart diagram for each metal.

Round 2

Reviewer 1 Report

My previous comments were responded properly, I have no other recommendations regarding the manuscript.